# Generalizable knowledge outweighs incidental details in prefrontal ensemble code over time

Mark D Morrissey[1,2], Nathan Insel[1], Kaori Takehara-Nishiuchi[1,2,3]*

[1]Department of Psychology, University or Toronto, Toronto, Canada; [2]Collaborative Program in Neuroscience, University of Toronto, Toronto, Canada; [3]Department of Cell and Systems Biology, University of Toronto, Toronto, Canada

**Abstract** Memories for recent experiences are rich in incidental detail, but with time the brain is thought to extract latent rules and structures common across past experiences. We show that over weeks following the acquisition of two distinct associative memories, neuron firing in the rat prelimbic prefrontal cortex (mPFC) became less selective for perceptual features unique to each association and, with an apparently different time-course, became more selective for common relational features. We further found that during exposure to a novel experimental context, memory expression and neuron selectivity for relational features immediately generalized to the new situation. These neural patterns offer a window into the network-level processes by which the mPFC develops a knowledge structure of the world that can be adaptively applied to new experiences.

*For correspondence: takehara@psych.utoronto.ca

**Competing interests:** The authors declare that no competing interests exist.

## Introduction

Knowledge about the world is thought to involve the statistical integration of correlations within and across many individual experiences (*Ghosh and Gilboa, 2014*; *McClelland et al., 1995*; *Preston and Eichenbaum, 2013*). While knowledge forms from experiences, over time memories of the experiences themselves often lose their contextual richness (*Furman et al., 2007*; *Sekeres et al., 2016*; *Winocur et al., 2007*). According to theories of systems memory consolidation, the process by which knowledge is formed involves the gradual reorganization of networks within interconnected brain regions that include the hippocampus and neocortex (*McClelland et al., 1995*; *Winocur et al., 2010*). One region that seems to be particularly involved in long-term memory is the medial prefrontal cortex (mPFC; *Frankland et al., 2004*; *Takashima et al., 2006*; *Takehara et al., 2003*; *Takehara-Nishiuchi and McNaughton, 2008*). The mPFC is necessary for learning new goals and behaviors when this learning depends on pre-existing knowledge (rodents: *Richards et al., 2014*; *Tse et al., 2011*; *Wang et al., 2012*; humans: *Ghosh et al., 2014*), and human imaging data indicate that the mPFC is activated when subjects use existing knowledge to encode new information (*van Kesteren et al., 2010*; *Kumaran, 2013*), make inferences (*Zeithamova and Preston, 2010*), and guide decision making (*Kumaran et al., 2009*). The question remains as to how this role is supported by changes in neural signaling that take place throughout learning and subsequent consolidation. Based on the hypothesis that the mPFC provides the brain with schematic knowledge—*i.e.*, a framework of abstract associations about environment-behavior relationships—we predict that mPFC neuron ensembles build representations of correlations common to multiple experiences over a time period that systems consolidation is known to take place; furthermore, we expect that information for these common relationships is disproportionately represented relative to context-specific information.

**eLife digest** Many events in our lives resemble experiences we have had before, without being identical to them. Whenever you attend a party, for example, you may well take along a gift, such as a bottle of wine or a box of chocolates, but the gift will differ on each occasion. Psychologists believe that as our memories for such events become older, the incidental details unique to each event (such as the identity of the gift) are mostly forgotten. However, the common underlying patterns (what parties are like in general) are retained. This allows us to accumulate knowledge to guide our behavior in similar situations in the future.

Studies in rodents and people have shown that a region of the brain called the medial prefrontal cortex stores long-term memories about experiences. But to what extent do neurons in this region represent abstract generalized knowledge as opposed to the specific incidental details?

To find out, Morrissey et al. used hair-thin electrodes to record the activity of hundreds of cells in the medial prefrontal cortex as rats performed a learning and memory task. The rats learned that either a tone or a light signaled the delivery of a mild electric shock. Initially, cells in the medial prefrontal cortex responded differently to the tone and to the light. However, after three weeks, the cells began to show similar responses to both stimuli.

The medial prefrontal cortex activity had thus transitioned from representing incidental details (tone versus light) to representing abstract relationships (stimulus predicts shock). This may relate to how the brain extracts commonality across experiences. A lingering question is how cells in the medial prefrontal cortex become selective for abstract relationships. We know that memories are reactivated during sleep. Therefore, one possibility is that combined reactivation of different experiences selectively strengthens memories for any features common to those experiences.

Here we examine how two different memories with overlapping associative structures are coded by neuron populations in the mPFC of rats, and how these codes change over time. The two memories both rely on a trace eyeblink conditioning procedure, in which a neutral stimulus (CS) is paired with eyelid stimulation (US) with a stimulus-free interval between CS offset and US onset. The retrieval of this memory initially depends on the hippocampus, but over the course of two to four weeks becomes dependent on the prelimbic region of the mPFC (*Takehara et al., 2003*; *Takehara-Nishiuchi et al., 2006*). Over this same time, mPFC neuron ensembles develop selective firing patterns for the acquired CS-US associations and maintain stable representation thereafter, a change that takes place with or without continued conditioning (*Hattori et al., 2014*; *Takehara-Nishiuchi and McNaughton, 2008*). These observations make trace eyeblink conditioning ideal for examining the evolution of the selectivity of mPFC ensemble activity across time. We find that the development of mPFC population codes for common task features involves bi-directional changes in selectivity for relational versus physical stimulus features over weeks after learning.

## Results

### Rats learned two associative memories sharing a common stimulus relationship

It was first important to establish that rats were capable of learning the associations of both visual and auditory CSs, while at the same time learning to discriminate CS-US trials from control trials in which the CSs were presented alone. Four rats underwent daily trace eyeblink conditioning with each session divided into two epochs (*Figure 1A*; *Takehara-Nishiuchi and McNaughton, 2008*). Within each epoch, 20 control trials were initially presented in which the auditory or visual CS was presented alone, these were then followed by 80 trials in which the CS was paired with the US. The trials were separated by an interval that was pseudorandomized across trials and ranged from 20 to 40 s. The CS-alone trials were critical to establish neural selectivity for the CS-US relational structure. Over the course of approximately 12 conditioning sessions, rats developed anticipatory blinking responses (conditioned responses, CRs) that peaked near the expected onset of the US; this anticipatory behavior was specific for CS-US paired trials and was not observed in CS-alone trials

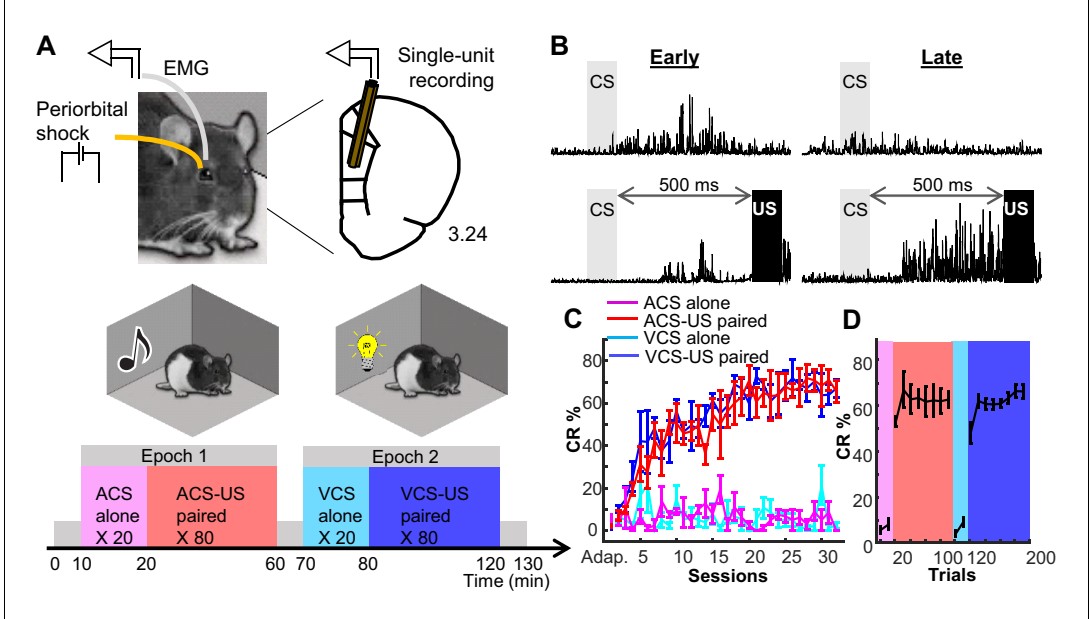

**Figure 1.** Rats formed two associative memories that differed in incidental, physical features. (**A**) Rats were implanted with two eyelid wires to deliver mild electrical current (US) and record anticipatory blinking activity (CRs). Single neural activity was collected from the prelimbic region of the mPFC from the first day of conditioning. Conditioning took place in two chambers, in which either a tone (ACS) or light (VCS) was presented alone (CS alone, 20 trials) or preceding the US by 500 ms (CS-US, 80 trials). (**B**) Averaged EMG amplitude during Early (left) and Late learning (right) during CS-alone trials (top) and CS-US paired trials (bottom). (**C**) CR expression (CR%; ±SEM) increased over days for both ACS-US (red) and VCS-US (blue) conditions, but not during the respective CS-alone conditions (magenta, turquoise). (**D**) Within sessions in which rats reached asymptotic responding, CR% showed an abrupt transition upon the shift from the block of CS-alone trials (magenta; turquoise) to the block of CS-US paired trials (red; blue).

The following figure supplement is available for figure 1:

**Figure supplement 1.** Behavioral performance in two blocks of trace eyeblink conditioning.

(*Figure 1B*). Importantly, the increased frequency of eyeblink responses (CR%) was only observed during the period before US onset (post-CS phase; *Figure 1C*; for CR% in individual rats, see *Figure 1—figure supplement 1A*), but not during the period before CS onset (pre-CS phase; *Figure 1—figure supplement 1B*). Three-way repeated measures ANOVA revealed a significant phase × trial type × session interaction, $F_{75,468} = 235.66$, p<0.001). In CS-US paired trials, CR% during the post-CS phase became greater across sessions, but CR% during pre-CS phase did not (follow-up two-way repeated measure ANOVA, phase × session interaction, $F_{25,156} = 7.931$, p<0.001). In contrast, in CS-alone trials, CR% did not change across sessions or differ between the pre- and post-CS phases (phase × session interaction, $F_{25,156} = 0.684$, p=0.867). A nonparametric test also showed the significant difference in CR% across four trial types during the post-CS phase (Friedman test, $\chi^2_3$ = 327.36, p<0.001). Once performance reached asymptote, CR% showed an abrupt transition between CS-alone and CS-US trials in each session (*Figure 1D*). This discontinuity suggested that the learning process involved successful encoding of the distinct temporal context between earlier and later trials; *i.e.*, that CS-alone trials at the start of the trial block did not simply extinguish associations acquired on previous days. Thus, the present behavioral protocol enabled the rats to acquire two associative memories (ACS-US and VCS-US association) that shared a common, relational feature (*i.e.*, the CS-US association), but differed in a discrete, physical feature (*i.e.*, the sensory modality of CS). They also acquired the temporal context that initial trials within each epoch would not be paired with the US.

## Prefrontal ensembles form codes selective for physical and relational features of memories

To track neuron activity, and thereby measure the selectivity of the neural population, we extracellularly recorded action potentials of neurons in the prelimbic region of the mPFC from the first day of learning (10–35 neurons per day; *Figure 2—figure supplement 1A–C*). Recordings were performed with a chronically implanted microdrive (*Kloosterman et al., 2009*) containing 14 independently-movable, four-channel electrodes ('tetrodes', [*Wilson and McNaughton, 1993*]). The movement of each tetrode was minimized to sample the activity of neurons located in a comparable part of the prelimbic region throughout the one-month period of daily recording. Our data, therefore, consist of some neurons sampled repeatedly across days and others sampled once (*Figure 2—figure supplement 1D*). During learning, the neuron ensemble in the mPFC showed different firing rate changes to the auditory and visual CS (*Figure 2A*; see also *Figure 2—figure supplement 2*, for the remaining three rats), and also distinguished between the auditory CS presented alone and the auditory CS paired with the US. Differential firing to the two CSs was evident during the CS (*Figure 2B*; permutation tests on the similarity of ensemble patterns for ACS alone and ACS-US paired vs. that for ACS-US paired and VCS-US paired, p=0.001 and 0.002 in two 50-ms bins during the CS) while differential patterns between CS-alone and CS-US trials were detectable during the interval between the CS and US (in 3 out of 8 bins, p<0.002). By the third week after learning, at which point the behavioral expression of the CS-US association is known to depend on the mPFC (*Takehara et al., 2003*; *Takehara-Nishiuchi et al., 2006*), the mPFC ensemble differentiated between CS-alone trials and CS-US trials more than it differentiated between visual and auditory CS trials (*Figure 2A*, *Figure 2—figure supplement 2*). During the CS, the similarity of neuron firing patterns for two trial types with the same CS was no longer different from that for trial types with different CSs (*Figure 2B*; permutation test, p=0.099, 0.981). During the CS-US interval, however, the similarity of neuron firing patterns became higher for trials with the shared stimulus relationship than for those without (in 6 out of 8 bins, p<0.001). Similar patterns were also observed in changes of firing rate evoked by the CS (*Figure 2—figure supplement 3*). Whereas during learning, the ensemble patterns of CS-evoked firings were similar between CS-alone and CS-US paired trials, they became more similar for ACS-US and VCS-US paired trials during the third post-learning week. At a finer timescale, within a single session of the third post-learning week, the ensemble activity rapidly evolved into a new, statistically uncorrelated pattern within the first ten ACS-US paired trials (*Figure 2C*), which was tightly coupled with the time course over which the rats increased the frequency of CR expression at the beginning of ACS-US paired block (*Figure 1D*). Upon the transition from the epoch with the ACS to another with the VCS, the ensemble similarity drastically dropped but went up again as the block of VCS-US pairings began. In contrast, during learning, the degree of ensemble differentiation across the four conditions appeared to be smaller, and it took a greater number of trials until the ensemble pattern evolved into a new pattern upon the block shift (*Figure 2—figure supplement 4*). Collectively, these observations suggest that initially, the mPFC ensemble encoded both the physical as well as relational features of the stimuli to a comparable degree; however, after extensive experience, the mPFC ensemble code became less sensitive to physical (sensory) features unique to each memory and more sensitive to their common, relational feature.

## Enhanced selectivity of relational features parallels reduced selectivity for physical features across time

To unpack the observed difference in neuron selectivity between learning and post-learning periods, we tracked mPFC ensemble selectivity for relational (CS-alone versus CS-US trials) and physical (ACS versus VCS trials) features over five successive stages of learning. For these analyses, we used a machine classifier to establish the degree to which neuron populations differentiated—i.e., could successfully distinguish—trials of each condition. Conditioning sessions were divided into five learning stages separately for each rat based on CR rate (*Figure 1—figure supplement 1A*). The stages were: (1) Before Learning, when CR% was less than 30, (2) Learning, during which the rate of CRs progress to an asymptotic level, and (3–5) Post 1W, 2W, and 3W, which correspond to the first, second, and third week after CR expression reached asymptote. In each stage, selectivity of the mPFC ensemble activity for relational and physical features was quantified by applying a Support Vector

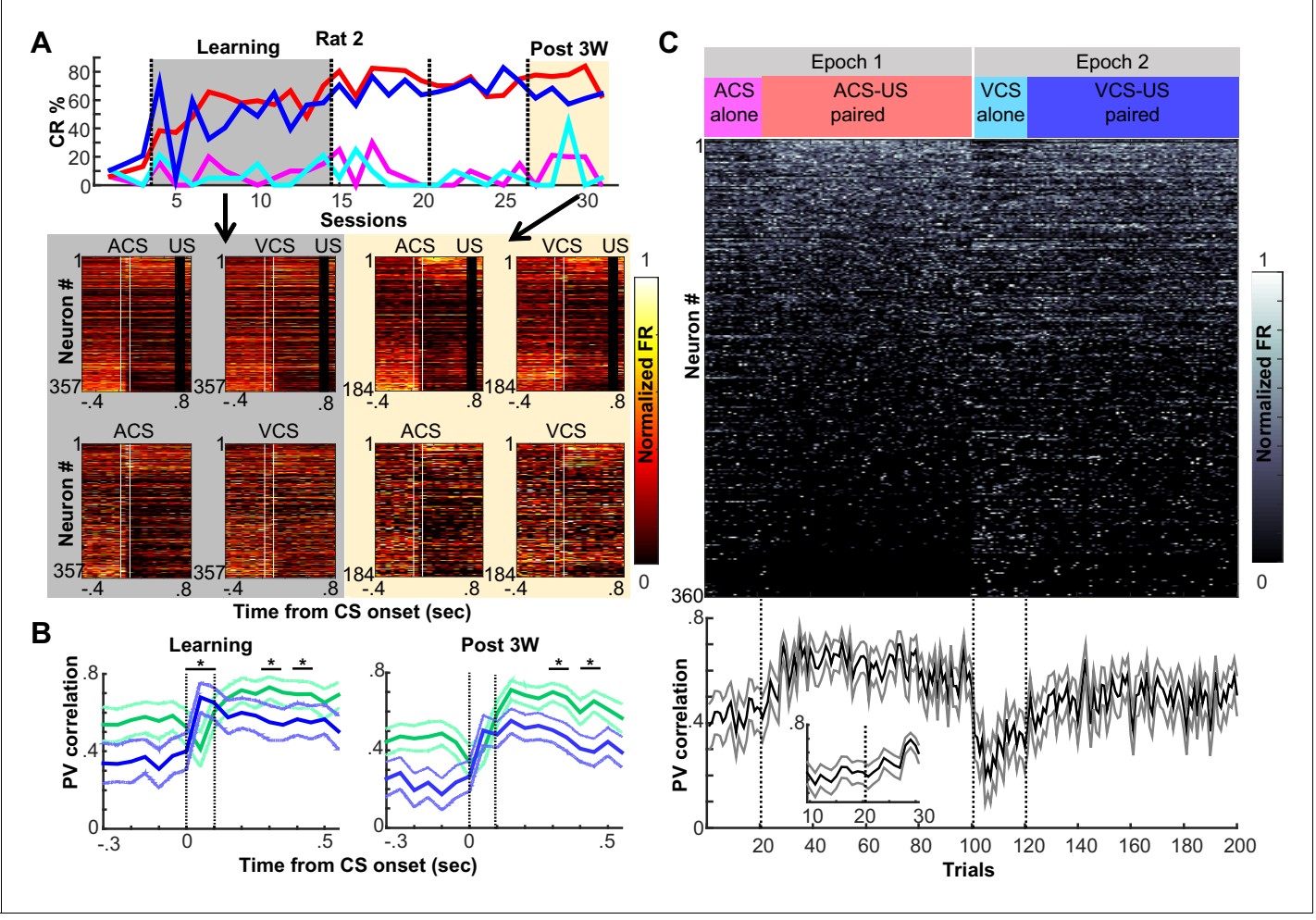

**Figure 2.** Prefrontal ensemble codes in well-trained rats were selective for relational features. (A) Example behavior from a single rat. Based on the percentage of trials exhibiting a conditioned response (CR%) and days since asymptotic responding, sessions were divided into five successive stages (vertical lines). Representative pseudocolor plots show normalized firing rate of neurons recorded from the same rat during the learning period (gray) and during the third week after learning (yellow). Top left, ACS-US paired; top right, VCS-US paired; bottom left, ACS alone; bottom right, VCS alone. Neurons were sorted based on the ACS-induced firing rate change during ACS-US pairings from the largest increase (Neuron #1) to the largest decrease (Neuron #357 or 184). During learning, ensemble activity during ACS-US pairings was similar to that during ACS alone presentations and VCS-US pairings; however, during the third post-learning week, it became more similar to that during VCS-US pairings than ACS alone presentations. White lines indicate CS onset and offset, black bars mask US artifacts. (B) During learning (left), binned firing rates of neuron ensembles (PV) during the CS (two vertical lines) were more similar for two conditions with the same modality of CS (ACS-US paired and ACS alone, blue, r ± 95% confidence intervals) than two conditions with the same stimulus relationships (ACS-US paired and VCS-US paired, green). Their similarity became comparable between two condition pairs during subsequent CS-US intervals. During the third post-learning week (right), PV became more similar for two conditions with the shared stimulus relationship than those with the shared CS modality. * indicates p<0.05/10 in random permutation tests. (C) Trial-by-trial display of PVs during CS-US intervals in the third post-learning week. Neurons from four rats were sorted based on the firing rate during ACS-US paired trials from the highest (Neuron #1) to the lowest (Neuron #360)]. A 'template' PV was constructed as averaged PVs across 10–80th ACS-US paired trials. The correlation coefficient between the template and PV in each trial (±95% confidence interval) rapidly increased within the first ten ACS-US paired trials (inset). Upon the transition from the epoch with the ACS to the next epoch with the VCS, it abruptly decreased but increased again when the VCS-US pairings began.

The following figure supplements are available for figure 2:

**Figure supplement 1.** Histology and single unit isolation.

**Figure supplement 2.** Prefrontal ensemble activity during four conditions with different relational and physical stimulus features.

**Figure supplement 3.** CS-evoked firing patterns during four conditions with different relational and physical stimulus features.

*Figure 2 continued on next page*

*Figure 2 continued*

**Figure supplement 4.** Trial-by-trial changes in the similarity of ensemble activity during learning.

Machine (SVM) classifier to binned firing rates during intervals between the CS and US in four conditions. Better performance of the classifier reflects higher selectivity of neuron population firings for the relational and physical features that differentiate the four conditions. The classification accuracy greatly varied across five stages of learning (*Figure 3A*; one-way ANOVA, $F_{4,95} = 11.34$, p<0.001). Examination of the specific classification errors (the 'confusion matrix', *Figure 3B*) showed that in the Before Learning and Learning stages, the majority of inaccurate classifications resulted from errors in discriminating CS alone trials from CS-US paired trials. In contrast, across the post-learning weeks, the classifier made more errors in discriminating ACS trials from VCS trials, suggesting that across time prefrontal ensemble activity appeared to become more sensitive to the paired vs. unpaired trial blocks and less sensitive to stimulus modalities. This was confirmed by applying the SVM method to make binary discriminations in the relational (CS alone vs. CS-US trial blocks) and physical (ACS vs. VCS trial blocks) dimensions: classification accuracy for the relational feature significantly improved across stages (*Figure 3C*; one-way ANOVA, $F_{4,95} = 9.99$, p<0.001); while classification accuracy for the physical feature steadily decreased (one-way ANOVA, $F_{4,95} = 3.77$, p=0.007). Notably, increased selectivity for the relational dimension took place primarily during first and second post-learning weeks (Before Learning versus Post 2, or 3W, Learning versus Post 1, 2, or 3W, Tukey HSD, p<0.05), whereas selectivity for the physical dimension decreased during the third post-learning week (Post 3W versus Before Learning, Tukey HSD, p<0.05). These results suggest that the process by which the mPFC extracts knowledge about environmental structure across time, as reflected in the increased coding for stimulus relationships, may be independent from its declining sensitivity to sensory features of the stimuli.

### Time-dependent changes in the selectivity for relational and physical features were driven by independent changes in single neuron firing properties

We next evaluated how the ensemble changes corresponded to changes of feature selectivity within single neurons. Among 2101 neurons, 65.1% significantly changed firing rates during the CS or CS-US interval relative to inter-trial intervals in at least one of four conditions; this proportion did not appear to change between learning and over-training periods (*Figure 4—figure supplement 1*). Of these rate-changing neurons, 10.5% showed selectivity for the relational features of the stimulus ('Relational'), meaning they exhibited a significantly different response pattern during CS-US paired trials compared with CS-alone trials, regardless of the modality of the CS (alpha = 0.05, random permutation test with 1000 samples; *Figure 4*). A separate set of rate-changing neurons, 13.7%, were selective for the stimulus modality (*i.e.* the physical features of the CS, 'Physical'), meaning firing rates differed between ACS and VCS trials, regardless of whether it was a CS-alone or CS-US paired trial. More neurons (17.6%) were selective for both relational and physical features (Conjunctive). The remaining neurons showed the same response patterns in all four conditions.

To quantify learning stage-dependent changes in the feature selectivity of single neurons, we computed two independent properties, the magnitude and the consistency of differential firing rates between conditions (*Figure 5A*). Over the five learning stages, the magnitude of differential firing for the relational dimension increased during the CS-US interval, as measured by mean ranks of a 'Differentiation index' (0-none to 1-strongest; *Figure 5B*, Kruskal-Wallis test, $\chi^2_4 = 19.37$, p<0.001). Magnitude of differential firing became significantly higher during the Post 1 and 3W stages compared with the Before Learning stage (Wilcoxon rank sum test, p<0.001,=0.020, and 0.001 for Post 1, 2, and 3W, respectively). In contrast, the magnitude of differential firing for the physical feature did not significantly change across the stages ($\chi^2_4 = 6.14$, p=0.189) nor were there changes in the proportion of neurons with a high differentiation index for the relational or physical feature (*Figure 5C*).

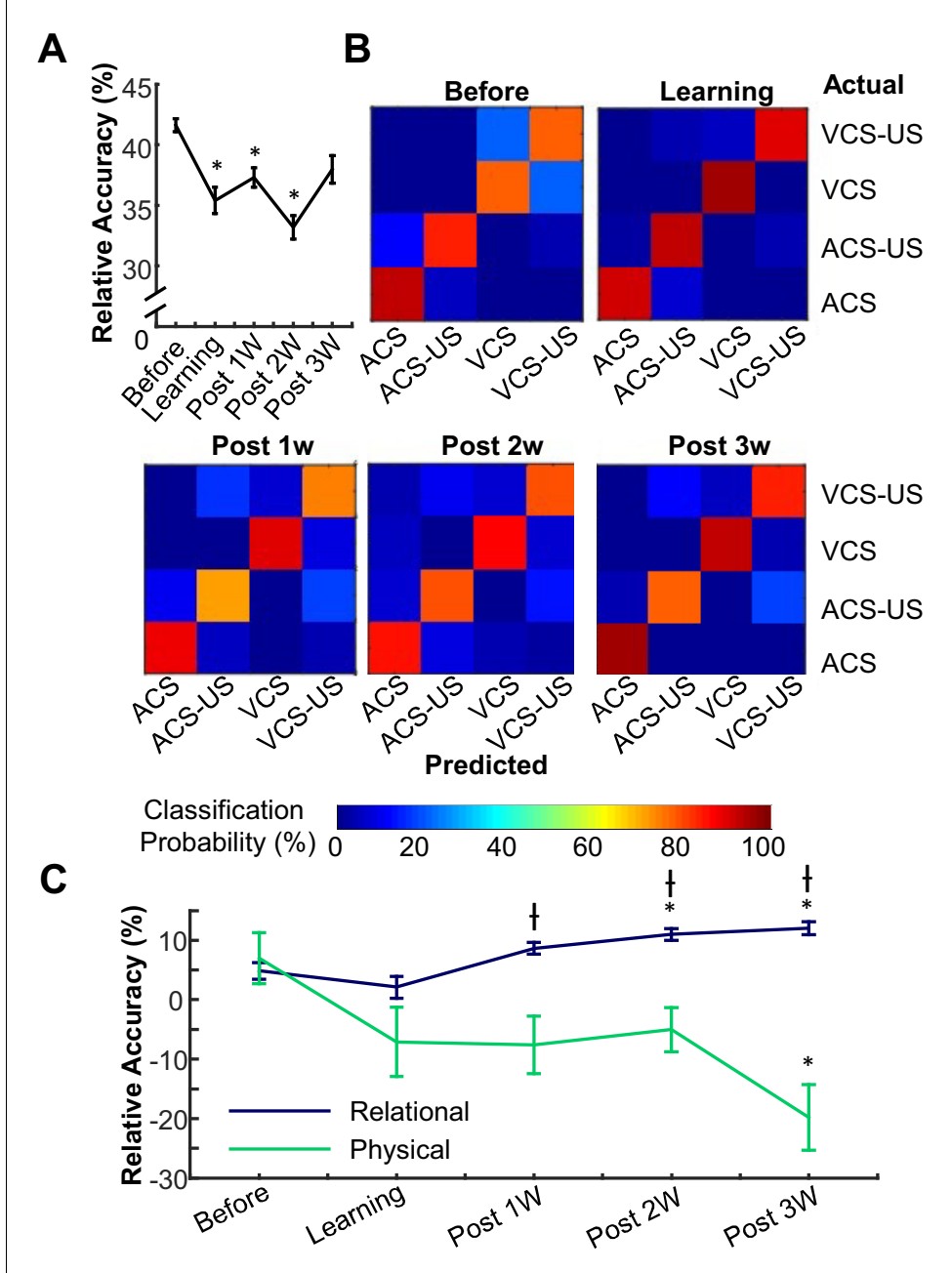

**Figure 3.** Prefrontal neuron ensembles became more selective for relational and less selective for physical features across learning stages. (**A**) The relative accuracy of Support Vector Machine (SVM), trial type decoding (raw accuracy minus accuracy at chance) was separately calculated in each of five successive stages of learning. Decoding accuracy greatly fluctuated across stages. (**B**) Representative confusion matrices from SVM showing decoding accuracy between trial types. Each square in each 4 × 4 matrix shows the probability (color) that a trial of one type (rows) was classified as another type (columns). Hotter colors along diagonal illustrate the high accuracy of the classifier. During the Before learning stage, most classification errors were between CS-alone and CS-US trials, illustrated by lighter blue squares within the upper left and lower right quadrants. In the later stages, errors became more common between ACS-US and VCS-US trials. (**C**) Average SVM accuracy for binary discrimination along relational (navy) and physical (green) dimensions. Curves emphasize increased information about stimulus relationships and decreased perceptional information in the ensemble. (* indicates $p < 0.05$ in comparison with Before Learning, † shows $p < 0.05$ in comparison with Learning).

The following figure supplement is available for figure 3:

*Figure 3 continued on next page*

*Figure 3 continued*

**Figure supplement 1.** Parameters that affected the ability of support vector machine classifier to decode conditions based on prefrontal ensemble activity.

Differences were also observed with respect to the consistency of differential activity between trial types, as measured by mutual information. Mean ranks of mutual information for the physical feature significantly decreased across the five learning stages (*Figure 5D*, $\chi^2_4$ = 12.06, p=0.017), but those for the relational feature did not ($\chi^2_4$ = 2.27, p=0.686). The consistency of differential firing became significantly lower during the Post 1 and 3W stages compared with the Before Learning stage (Wilcoxon rank sum test, p=0.003, 0.016, and 0.001 for Post 1, 2, and 3W, respectively). Similarly, there was a trend toward a decline across stages in the proportion of neurons with a significant mutual information value for the physical features (random permutation test, p<0.05; binomial test, Before vs. Post 3W, p<0.1; *Figure 5E*), while changes in this proportion were not observed for the relational feature. These findings suggest that the increase in ensemble selectivity for relational information is mediated by increases in the magnitude of differential firing, whereas reduced selectivity for perceptual information involved changes in the consistency of differential firing.

## Generalization of an existing ensemble code to a new environmental context

Knowledge is useful when it can be applied to new situations. To examine whether relational coding generalized to a novel situation, three rats were trained for approximately 30 sessions in the paradigm described above, and were then presented with the same structure of 20 CS-alone trials and 80 CS-US trials, using only the auditory CS, in the old chamber (Box 1) and in a new conditioning chamber (Box 2; *Figure 6A*). The new chamber differed from the old chamber in the visual appearance of the walls, illumination, and texture of the floor. The rats immediately responded with CRs to

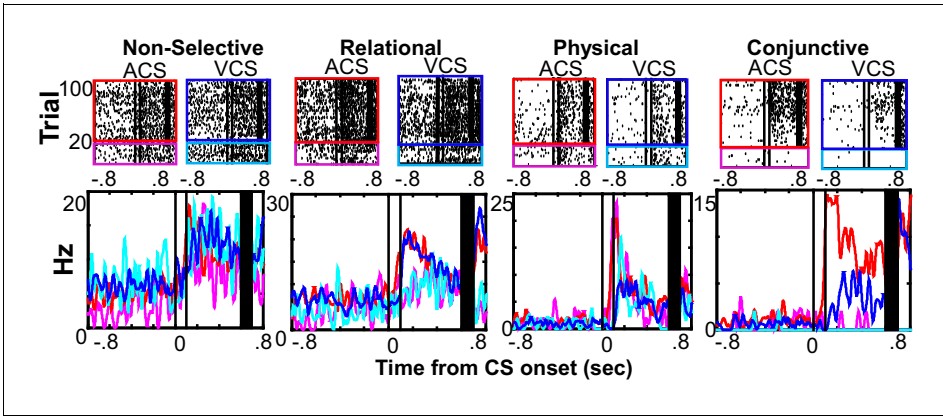

**Figure 4.** Single prefrontal neurons showed selectivity for the different features of the memory. Representative raster plots and peri-stimulus time histograms (1 ms bins, smoothed with a 50 ms Hanning window) for each of four conditions (the presentation of auditory CS alone, magenta; pairings of the auditory CS and US, red; the presentation of visual CS alone, turquois; pairings of the visual CS and US, blue). Although some neurons showed the same CS-evoked firing patterns across four conditions (Non-selective), others were found with firing rate changes dependent on a relational feature (Relational, rates in CS-US paired trials differed from rates in CS-alone trials), a physical feature (Physical, rates in trials with the ACS differed from those with the VCS), or their conjunction (Conjunction, rates in one condition differed from the other conditions). Two black lines indicate CS onset and offset, and black bars mask the artifact induced by the US.

The following figure supplement is available for figure 4:

**Figure supplement 1.** The proportion of neurons responding to the CS in each learning stage.

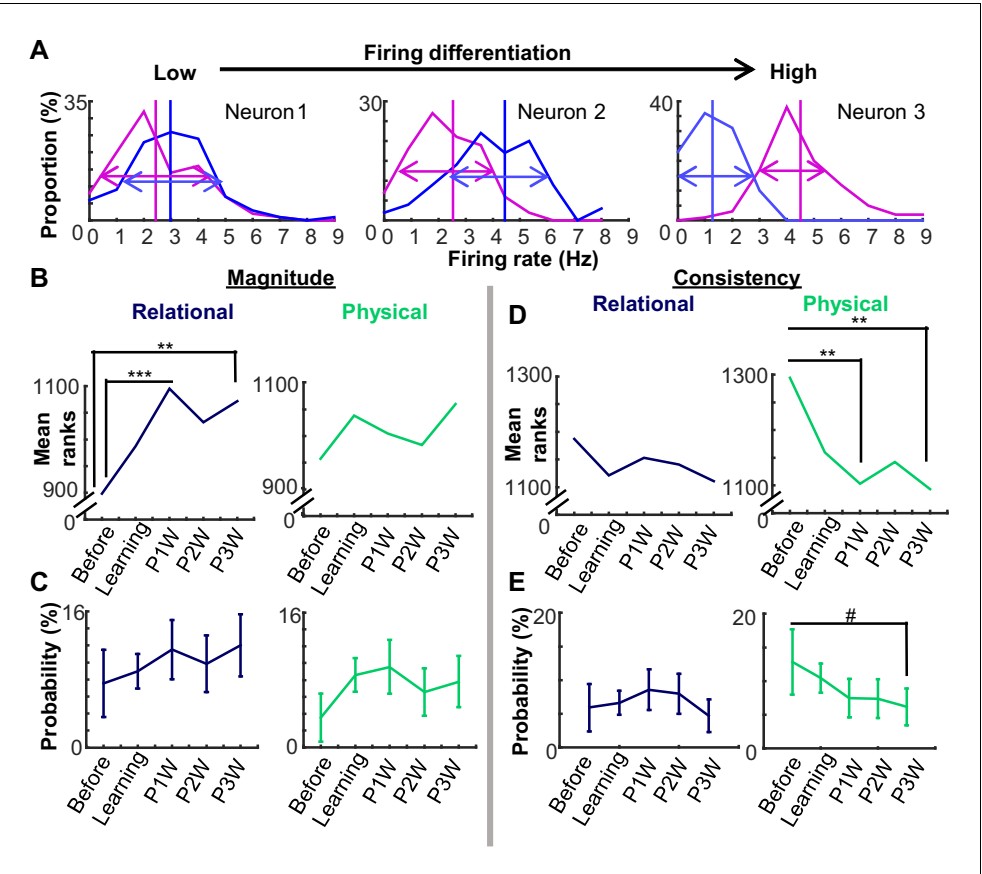

**Figure 5.** Changes in neural coding over time could be decomposed into changes in differential firing magnitude vs. consistency. (**A**) Three examples of firing differentiation of a neuron between trials with the auditory conditioned stimulus (CS, magenta) and those with the visual CS (blue). A neuron is more 'selective' for the physical feature, if it has a greater difference in the mean firing rate (vertical lines) between two conditions (Magnitude) and a smaller variance in trial-by-trial firing rates in each condition (arrows; Consistency). (**B**) Mean ranks of the differential index between CS-alone trials and CS-US paired trials (Relational, left) increased during weeks after learning, whereas that between the auditory and visual CS trials (Physical, right) did not. (**C**) The proportion of neurons (±95% confidence intervals) with high differentiation index for relational or physical feature did not change across the stages. (**D**) Across five stages, mean ranks of mutual information for the relational feature did not change. In contrast, mean ranks of mutual information for the physical feature decreased over the stages. (**E**) The proportion of neurons (±95% confidence intervals) with significantly high mutual information for the physical feature showed a trend toward decreasing across stages. (Symbols: #, **, and *** indicate p<0.01, 0.01, and 0.001, respectively).

the auditory CS-US pairings in the new chamber (*Figure 6B*). Prefrontal neurons exhibited similar firing patterns during CS-US pairings in the two chambers, while also maintaining clear differentiation between the CS-US pairings and CS-alone trials within each chamber (*Figure 6C*). Similarly, population decoding analysis revealed weak selectivity of neuron ensemble activity for the conditioning chamber but high selectivity for CS alone vs. CS-US trials (*Figure 6D,E*): the classifier made many errors discriminating CS-US pairings in Box 1 from those in Box 2 while making few errors discriminating the CS alone trials in Box 1 and Box 2. These data lead us to suggest that, like the neuron ensemble in the hippocampus (*Leutgeb et al., 2005*), the mPFC neuron ensemble is capable of generating separate codes for CS-alone trials in two different conditioning chambers, but that it actively assimilates previously-formed codes for stimulus associations to a novel context.

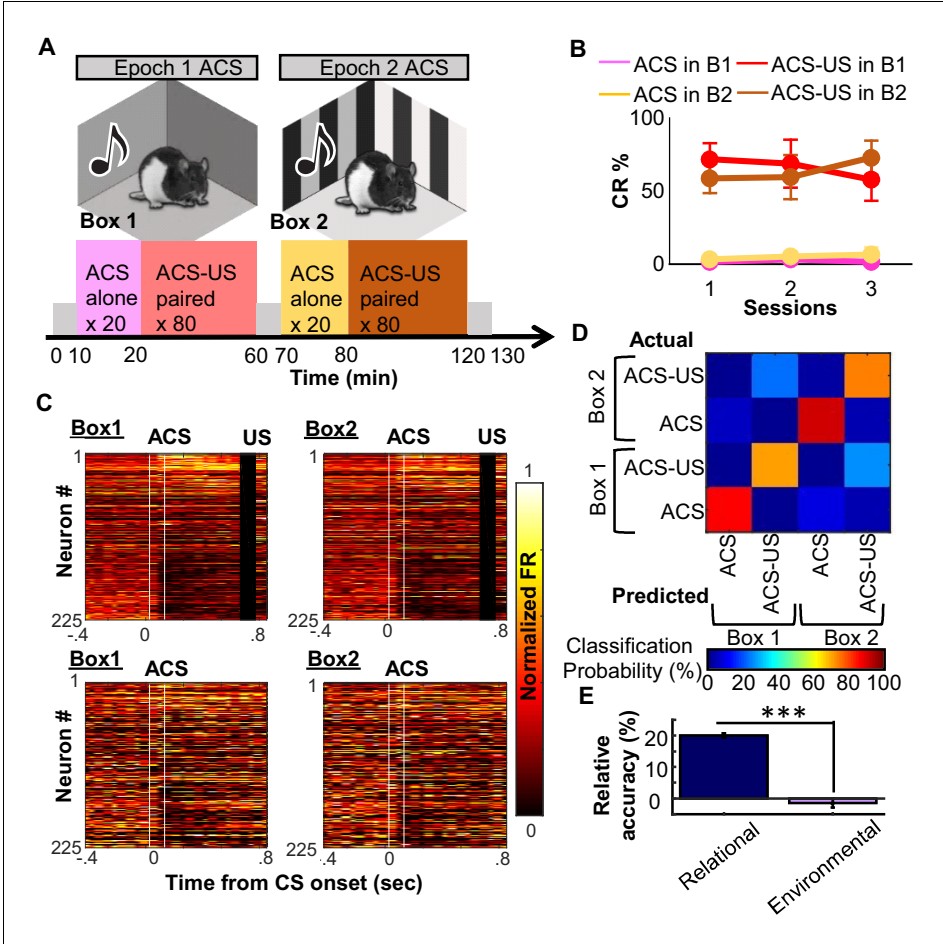

**Figure 6.** Use of existing ensemble code for CS-US relationship in a new context. (**A**) After ~30 days of conditioning three rats underwent three recording sessions in a familiar (Epoch1/Box1) and novel (Epoch2/Box2) environment. (**B**) Conditioned responses (CRs) were absent during CS-alone trials (Box1 pink, Box2 beige) and high during CS-US trials (Box1 red; Box2 brown), with no apparent differences between familiar and novel environment. (**C**) Pseudocolor plots show normalized firing rate of 225 neurons (sampled across three rats) during four conditions (top left, ACS-US in Box1; top right, ACS-US in Box2; bottom left, ACS-alone in Box1; bottom right, ACS-alone in Box 2). Neurons were sorted based on ACS-evoked firing rate change during the Box1, ACS-US pairings (top: largest increase in neuron #1, bottom: largest decrease in neuron # 225). Ensemble activity during ACS-US pairings in Box 2 was similar to that during ACS-US in Box 1, but not to ACS alone trials. (**D**) The confusion matrix from an SVM classifier of trial type. Most errors were misclassifications of Box1 versus Box2 ACS-US trials. (**E**) Relative decoding accuracy (raw minus accuracy at chance ±SEM across 20 runs) between ACS-US versus ACS alone trials and Box1 versus Box2 trials. *** indicates p<0.001.

## Discussion

Theories of systems consolidation posit that the neocortex gradually discovers common latent rules and structures from multiple past experiences and builds semantic knowledge of the external world (*McClelland et al., 1995*; *O'Reilly et al., 2014*). We show that, after learning, there is a gradual refinement of prefrontal neuron selectivity that may be a direct neuronal analog for this knowledge development process. Over a one-month period of repeated exposures to two similar experiences, mPFC ensemble activity gradually becomes more sensitive to their latent, relational variables; meanwhile, over time, information about perceptual, physical features of the environment is lost within this ensemble. Importantly, the selectivity for the physical features was weakened *after*, but not during learning, supporting the view that it underlies the mPFC's involvement in the extraction of commonalities from previous experiences (*Richards et al., 2014*; *Tse et al., 2011*; *Wang et al., 2012*),

rather than the learning of two stimulus associations. These results speak directly to a long-standing question in the field of whether the formation of generalized, or schematized memory is merely a product of the network 'forgetting' incidental features. Quite to the contrary, population analyses revealed different time-courses in the development of coding for relational features compared with the loss of selectivity for physical features. Examinations of single neuron activity also revealed that the former appears to rely more on increased magnitude of firing rate differences over weeks, whereas the latter appears to involve decreased reliability of differential firing. The presently observed prefrontal neuron ensemble changes are likely to be part of the physiological basis of the mPFC's contribution to an animal's ability to construct, maintain, and update an associative knowledge structure used to behave adaptively in familiar and novel environments (*Ghosh and Gilboa, 2014*; *van Kesteren et al., 2010*; *Preston and Eichenbaum, 2013*; *Richards et al., 2014*; *Tse et al., 2011*; *Wang et al., 2012*).

The stronger selectivity for relational over physical features is consistent with the previously reported selectivity of prefrontal neurons for rules (*Rich and Shapiro, 2009*; *Wallis et al., 2001*) or categories (*Freedman et al., 2001*) in well-trained animals. Our data reveal that this feature selectivity is not due to the innate inability of the mPFC ensemble to encode incidental features. In fact, during learning, the mPFC ensemble firing differentiated the physical features of the experiences to a comparable degree to the relational features (*Figures 2B* and *3C*, see also, *Hyman et al., 2012*; *Ma et al., 2016*). Furthermore, although the ensemble differentiation between ACS-US and VCS-US pairings decreased, neural responses to the two CSs remained highly differentiated in the CS-alone condition (*Figure 3B*). This observation is not consistent with a view that the weakened selectivity for the physical feature reflects a form of learned equivalence between the stimuli, arising from their equivalent associated outcome (*Honey and Hall, 1989*, *1991*; *Iordanova et al., 2007*; *Miller and Dollard, 1941*). Rather, the present findings support a view of the building of a new, high-level representation that takes into account the temporal context within which the stimuli are presented.

A question that remains unanswered is the type and range of experiences over which the mPFC network is capable of extracting commonalities. In the present study, both the temporal structure of the CS-US pairings and the outcome itself were common between the visual and auditory CS-US trials. Therefore, it remains unclear whether mPFC neurons only encode the predicted outcome, rather than the more abstract associative, temporal relationship between the stimuli. Results from other studies suggest that the mPFC can discriminate between situations with the same outcomes within the same environment, if only the rules for negotiating the environment differ (*Rich and Shapiro, 2009*; *Durstewitz et al., 2010*). Thus, although the present study did not explicitly alter the outcome, it seems reasonable to suppose that behavioral savings between paradigms with overlapping rules or temporal structure, even when the outcomes or unconditioned stimuli themselves are altered, will rely on representations within the mPFC that develop over consolidation, as described presently.

Is the observed change in the selectivity of prefrontal neurons a product of time passage after learning or repeated conditionings? Neural selectivity for the CS-US relationship (the relational feature) is no less a product of time than experience, because it becomes strengthened with or without repeated daily conditionings after learning (*Hattori et al., 2014*; *Takehara-Nishiuchi and McNaughton, 2008*). These observations, however, do not address whether the weakened selectivity for physical features requires continued conditioning. It is worth noting that the *loss* of selectivity may very well require no active process at all (see also, *Richards et al., 2014*), as it could be accounted for by the lack of reinforcement over time. In that sense, the repeated exposures in the present study may be working *against* the weakening of selectivity that we observed. Testing this point directly requires future studies which monitor the selectivity of mPFC neurons over time with manipulations of multiple variables that include the number of different 'CS-US' exemplars animals are exposed to, the similarity between exemplars, the temporal proximity between exposures, and elapsed time from the exposures.

When the animals were exposed to a new situation in which the same stimulus relationship took place in a novel environment, the mPFC immediately assimilated its code for the new situation to the existing, generalized code, without reverting to a code that also encodes incidental details (*Figure 6*). This is not due to the inability of mPFC ensemble to encode environmental features because it showed distinct firing patterns during two neutral experiences taking place in two different environments (*Figure 6D*; *Hyman et al., 2012*). This immediate transfer of an existing abstract code

may serve as a key computational basis for the assimilation of new information to a pre-existing knowledge structure (*Bartlett, 1932*). Experimental evidence from rodent behavioral studies and human imaging studies suggests that memory assimilation depends on the mPFC (*DeVito et al., 2010*; *Richards et al., 2014*; *Tse et al., 2011*; *Wang et al., 2012*), hippocampus (*Dusek and Eichenbaum, 1997*; *Iordanova et al., 2011*; *Tse et al., 2007*), and their interactions (*van Kesteren et al., 2010*; *Kumaran et al., 2009*; *Zeithamova et al., 2012*). Some theories posit that mPFC-hippocampal interactions during memory assimilation may be an extension of those during memory retrieval: where the mPFC selects a memory that is the most appropriate in a current context and sends top-down signals to the hippocampus to recover its contents (*Preston and Eichenbaum, 2013*). From a computational perspective, the initial selection process likely relies on pattern completion of mPFC ensemble activity from cues available in a current context (*Takehara-Nishiuchi and McNaughton, 2008*). This would activate downstream targets, including the rhinal cortices (*Paz et al., 2007*) that have connections with other neocortical regions as well as the hippocampus. The former may result in the recovery of a gist-like version of previous experiences (*Insel and Takehara-Nishiuchi, 2013*), whereas the latter may facilitate the acquisition of new information (*Bero et al., 2014*; *Preston and Eichenbaum, 2013*) by activating neurons bearing original memories (*McKenzie et al., 2013*; *Navawongse and Eichenbaum, 2013*; *Rajasethupathy et al., 2015*). This view unites two seemingly disparate engagements of the mPFC in initial learning and in the retrieval of consolidated memory into a common computation executed by the mPFC neuron ensemble.

In conclusion, our observations show the gradual development of the mPFC ensemble code for behaviorally relevant features common across multiple experiences, a process involving parallel modifications of two properties of single neuron firings across different time courses. This unique coding property of the mPFC may support its role in the formation, maintenance, and updating of associative knowledge structures that support flexible and adaptive behavior (*Ghosh and Gilboa, 2014*; *van Kesteren et al., 2010*; *Preston and Eichenbaum, 2013*; *Richards et al., 2014*; *Tse et al., 2011*; *Wang et al., 2012*).

## Materials and methods

### Animals

All experiments were performed on four male Long-Evans rats (Charles River Laboratories, St. Constant, QC, Canada) between 16–25 weeks old at the time of surgery. Rats were housed individually in Plexiglass cages and maintained on a reversed 12 hr light/dark cycle. Water and food were available *ad libitum*. All methods were approved by the Animal Care and Use Committee at the University of Toronto.

### Electrodes for single-unit recording

Tetrodes were made in-house by twisting together four 12 μm polyimide coated nichrome wires (Sandvik, Stockholm, Sweden) following our previous work (*Takehara-Nishiuchi and McNaughton, 2008*). To permit independently adjustable tetrode depths, each tetrode was housed inside a screw-operated microdrive. The complete microdrive-array consisted of a bundle of 14 microdrives, each guiding a tetrode, contained within a 3D printed plastic base (*Kloosterman et al., 2009*). The Microdrive-array also enclosed the Electrode Interface Board (EIB-54-Kopf, Neuralynx, Bozeman, MT, United States) to which all electrodes were connected and served as the interface between the recording and stimulating electrodes and the recording system. Prior to implantation, the impedance of the nichrome tetrode wires was reduced to ~250 kOhms by electroplating them with gold. Tetrodes were then drawn inside a stainless steel cannula (1.8 mm diameter) at the microdrive-array base and a small drop of sterilized mineral oil was added to ensure smooth movement of the tetrodes after implantation.

### Surgical procedures

Following guidelines set by the Institutional Animal Care Committee at the University of Toronto, all surgeries were conducted under aseptic conditions in a sterile surgical suite. For the chronic implantation of the microdrive array, rats were anesthetized with isoflurane (1–1.5% by volume in oxygen at

a flow rate of 1.5 L/min; Halocarbon Laboratories, River Edge, NJ, United States) and placed in a stereotaxic holder with the skull surface in the horizontal plane.

All tetrodes were targeted to the prelimbic region of the medial prefrontal cortex (PrL mPFC). The tetrode bundle was implanted with the same procedure as those used in our previous work (*Takehara-Nishiuchi and McNaughton, 2008*). A craniotomy was opened over the PrL mPFC at 3.2 mm anterior and 1.4 mm lateral to bregma and the dura matter removed. The microdrive array was then lowered at a 9.5° medial angle until the base made contact with the surface of the brain. The craniotomy was then sealed with Kwik-Sil (Stoelting, Kiel, WI, United States) and the array was held in place with self-curing dental acrylic (Lang Dental Manufacturing, Wheeling, IL, United States).

Immediately after the surgery, all tetrodes were lowered 1 mm into the brain. For the next 3–4 weeks, the rat was connected to the system each day to visualize the quality of activity and monitor movement of the tetrodes. Each tetrode was lowered slightly each day (75–125 μm) over the course of this 3–4 week period to target tetrodes tips to the PrL mPFC at 3.0–4.0 mm ventral from the brain surface. One tetrode was positioned superficially in the cortex (1 mm below brain surface) to serve as a reference electrode for single-unit activity. Once the recordings began, tetrode position was adjusted only as necessary to obtain good quality high yield recordings. This approach was necessary to sample the activity of neurons from comparable parts of the prelimbic region across five learning stages over a month. Tetrode position adjustments were only made after a given recording session providing ~24 hr for the tetrode to stabilize prior to the next recording.

## Behavioral paradigm and data acquisition

All rats experienced the same general experimental procedure. Beginning 3–4 weeks following microdrive array implantation, when stable single unit recordings were achieved and tetrodes were positioned within the PrL mPFC, rats were subjected to daily conditioning in the trace eyeblink conditioning paradigm.

Rats were placed in a large dark rectangular box, fitted with an LED light source and speaker. Within the box rats were enclosed in a square plexiglass container (20 × 20 × 25 cm), fitted with holes on one side to enable sound-waves from the speaker to enter the enclosure. The conditioned stimulus (CS) was presented for 100 ms and consisted of an auditory stimulus (85 dB, 2.5 kHz pure tone) or a visual stimulus (white LED light blinking at 50 Hz). The unconditioned stimulus (US) was a 100 ms mild electrical shock to the eyelid (100 Hz square pulse, 0.3–2.0 mA), and the intensity carefully monitored via webcam and adjusted to ensure a proper eyeblink/head turn response (*Morrissey et al., 2012*; *Tanninen et al., 2015*). The timing of CS and US presentation was controlled by a microcomputer (BasicX, Netmedia, Tucson, AZ, United States), and the US was generated by a stimulus isolator (ISO-Flex, A.M.P.I., Jerusalem, Israel).

Daily recording sessions consisted of two epochs of conditioning, each with 100 trials, separated by a 10 min rest period. Each epoch included 20 presentations of the CS alone, followed by 80 trials in which the CS was paired with the US, separated by a stimulus-free interval of 500 ms (see *Figure 1A*). The first epoch used only one of the two CS (*e.g.* auditory CS), and the second epoch used the other CS (*e.g.* visual CS), with the CS order and schedule pseudorandomized across days and across rats. This design provided four conditions for comparison: presentations of the auditory CS alone (ACS alone), pairings of auditory CS and US (ACS-US paired), presentations of the visual CS alone (VCS alone), and pairings of VCS and US (VCS-US paired). Before and after each epoch the rat was placed in a comfortable rest box separate from the conditioning box for 10 min.

Upon completion of the full conditioning procedure, several animals (n = 3), underwent a similar conditioning procedure over three days in which the conditioning environment was manipulated, but the same CS was used in two epochs. The rats underwent two epochs of 20 ACS alone trials and 80 ACS-US paired trials. One epoch was run in the same conditioning chamber as the previous 30+ days of conditioning (Box 1; a dark box with brown floors and plain walls), in the other epoch the conditioning took place in a box in which the visual and textile features were manipulated (Box 2; lit box, stripped walls, white floor). The epoch order was pseudorandomized across rats.

During the daily conditioning sessions, we simultaneously recorded action potentials from individual neurons in the prelimbic region of medial prefrontal cortex and electromyogram (EMG) activity from the eyelid. Action potentials were captured using the tetrode technique, which allows for recording the activity of many individual neurons per recording session (*Wilson and McNaughton, 1993*). Experimental rats were connected to the system through an Electrode Interface Board (EIB-

54-Kopf, Neuralynx, Bozeman, MT, United States) contained within the microdrive array fixed to the animal's head. The EIB was connected to a headstage (HS-54, Neuralynx, Bozeman, MT, United States), and signals were acquired through the Cheetah Data Acquisition System (Digital Lynx and Cheetah Software, Neuralynx, Bozeman, MT, United States). A threshold voltage was set at 40–50 mV, and if the voltage on any channel of a tetrode exceeded this threshold, activity was collected from all four channels of the tetrode. Spiking activity of single neurons was sampled for 1 ms at 32 kHz and signals were amplified and filtered between 600–6000 Hz. EMG activity was continuously sampled at 6108 Hz and filtered between 300–3000 Hz.

## Behavior analysis

Behavior was analyzed with the same procedures as those used in our previous studies (*Morrissey et al., 2012*; *Takehara-Nishiuchi and McNaughton, 2008*; *Tanninen et al., 2015*). The adaptive conditioned eyeblink response (CR) which represents the learning of the association between the conditioned stimulus (CS) and unconditioned stimulus (US) was assessed through the analysis of electromyogram (EMG) activity recorded from the upper left eye-lid muscle. Each trial was assessed offline with custom codes written in Matlab (Mathworks, Natick, MA, United States) for the presence of a CR. The CR was defined as a significant increase in eyelid EMG amplitude immediately before US onset. Specifically, EMG activity was sampled around the presentation of the CS in each trial and the instantaneous amplitude of the signal was calculated as the absolute value of the Hilbert transform of the signal (using the *hilbert* function in Matlab). For each trial, the average amplitude during a 300 ms period immediately before CS-presentation was defined as the Pre-Value. The averaged amplitude during a 200 ms period immediately before US-presentation was defined as the CR-Value in the post-CS phase, and the averaged amplitude during a 200 ms period around 0.9 s before CS onset was defined as the CR value in the pre-CS phase. A Threshold value was set as the averaged Pre-Value across trials plus two standard deviations. For a given trial, if the CR-Value exceeded the Pre-Value and the Threshold, that trial was classified as containing a CR. In some trials, the Pre-Value exceeded the Threshold value because the rats engaged in grooming, teeth grinding, or climbing immediately before CS onset (*Figure 1—figure supplement 1C*). These trials were classified as hyperactive and discarded. The proportion of these 'hyperactive' trials was typically ~5% and did not change across sessions in any of the trial types (*Figure 1—figure supplement 1D*). The ratio of trials containing a CR to the total number of valid trials within each of two conditions (CS alone, CS-US paired) represented the CR% for the condition for each epoch. CR% during four conditions was compared by using three-way repeated measures ANOVA with sessions, conditions, and phase as within-subjects factors as well as the Friedman test.

To assess changes in neuron activity across successive stages of learning, recording sessions were divided into five stages based on the frequency of CR expression. The criteria for stages were selected based on observations of general patterns of CR acquisition and expression across many animals. In the first few days of training, rats show very few trials in which they exhibit the CR, we define this period as the Before learning stage, *i.e.* before the animal has begun to associate the CS with the US. Once rats begin to form this association, the percentage of trials in which they exhibit a CR rapidly increases, but it can fluctuate greatly across days. We define this period as the Learning stage. Eventually the rats reach a point in which their responding plateaus and reaches asymptote, from this point on we generally observe small fluctuations in response rate across days but rarely see large deviations. This point defines the end of the Learning stage. All days beyond this point were defined in the Post-learning week stage. To operationally define these stages we set a threshold of responding. All days prior to the rat displaying the CR in 30% of trials are defined as the Before learning stage. All days following two consecutive days of the rat displaying the CR in 60% of trials are defined in weeks as three Post learning stages. All days in between Before learning and the beginning of the Post learning stage are defined as the Learning stage.

## Data preprocessing

Putative single neurons were isolated offline using a specialized software package in Matlab (KlustaKwik, author: K.D. Harris, Rutgers, The State University of New Jersey, Newark, NJ; MClust, author: D.A. Redish, University of Minnesota, Minneapolis, MN; Waveform Cutter, author: S.L. Cowen, University of Arizona, Tucson, AZ, United States). Both automatic spike-sorting and manual

sorting were used to assign each action potential to one of the neurons recorded simultaneously on one tetrode based on the relative amplitudes on the different tetrode channels and various other waveform parameters including peak/valley amplitudes, energy, and waveform principle components. The final result was a collection of time stamps associated with each action potential from a given neuron. Only neurons with <1% of inter-spike intervals distribution falling within a 2 ms refractory period were used in the final analysis. If a neuron did not show more than 1500 spikes during the entire recording session, it was removed from further analyses due to the insufficient number of spike waveforms to confidently judge if they were spikes recorded from a real neuron or noise. An individual neuron was defined as a unit that was well isolated from raw signals recorded on a tetrode. Because we minimized the movement of tetrodes across days, some units which appeared to belong to the same neuron were recorded across a few days (*Figure 2—figure supplement 1D*). We treated each of these units as a separate sample of a neuron. The total number of neurons is the summation of the number of isolated units across tetrodes, sessions, and rats (*Table 1*). Therefore, our data consist of some neurons sampled repeatedly across days and others sampled in one day. Because we were mainly interested in comparisons of ensemble selectivity across learning stages, having repeatedly sampled neurons across days was beneficial because it reduces the variability in sampled neurons across learning stages.

## Population firing rate matrix

To examine the similarity between firing rates of a population of neurons across four conditions (ACS-US paired, VCS-US paired, ACS alone, or VCS alone), we constructed four population firing rate matrices each of which contained the binned firing rate (50 ms) of all recorded neurons during a 1 s period around the CS onset (−400 to 600 ms) in one of the conditions. For each neuron, the firing rate in each bin was divided by its maximum firing rate across four conditions. We then sorted these neurons based on their change in firing rate during the CS-US interval relative to baseline during the ACS-US paired condition. To compare CS-evoked firing patterns across four conditions, raw firing rates of each neuron were converted to standard scores by using the mean and standard deviation of firing rates during a one-second period before CS onset.

To quantify the similarity of population firing rate matrices between two conditions, we calculated the Pearson correlation coefficient (r) between vectors of binned firing rates of two conditions that shared a relational feature (ACS-US paired and VCS-US paired) or a physical feature (ACS-US paired and ACS alone). To test whether r values for two condition pairs were significantly different from one another, we conducted random permutation tests. Trials were randomly assigned to either of two conditions in such a manner that the relative number of trials in each condition was held constant. The r value and its difference between two condition pairs were re-computed. This procedure was repeated 1000 times to construct sampling distributions. The difference in r values between two condition pairs was considered significant when it fell in the 0.0025% lower or upper tail of its corresponding distribution ($\alpha = 0.05/10$, adjusted for repetition across ten 50-bins covering from 0–500 ms after CS onset).

To examine trial-by-trial changes in ensemble similarity, we constructed population firing rate vectors which contained the firing rate of all neurons during intervals between CS offset and US onset in each of 200 trials in a session. We then defined a 'template' of ensemble activity for ACS-US pairings by averaging firing patterns across the 10-80th ACS-US paired trials. Pearson correlation coefficient (r) was calculated between the template and the firing vector of each trial.

## Population decoding analysis

To quantify the degree of selectivity of ensemble activity for physical and relational features of conditions, we examined how accurately a machine learning algorithm, Support Vector Machine (SVM) classifier (*Cortes and Vapnik, 1995*) could decode the conditions from binned firing rates of a neuron ensemble. Several studies have shown that the SVM classifier can be successful in decoding the identity of visual stimuli (*Nikolić et al., 2009*), the spatial position of a visual cue (*Astrand et al., 2014*), and the allocation of attention (*Tremblay et al., 2015*) from the activity of multiple single neuron firings. Moreover, the SVM classifier was shown to outperform several other commonly used classifiers (*Astrand et al., 2014*).

The SVM classifier produces a model from training data which then predicts the target values of test data given only the test data attributes. For the current study, the attributes were the normalized firing rates of a population of neurons in a trial of one of four conditions, and the target values were the condition from which they were sampled. The population firing vectors were constructed by concatenating the responses of a set of $N$ neurons on a trial from one of four conditions. Note that the neurons were recorded in separate sessions from four rats, and thus we ignored any correlated activity between neurons. Having simultaneous recordings, however, would most likely not have changed our conclusions since we were mainly interested in comparisons of relative classification ability across four conditions based on firing rate patterns immediately after CS presentations.

All algorithms were run in Matlab using the freely accessible LIBSVM library (*Chang and Lin, 2011*). The classifiers were trained with Radial basis function kernels. We first identified SVM parameters that maximized decoding accuracy by performing a grid search procedure (calculating decoding performance over a range of cost and gamma SVM parameters) for each set of training data. This was done by using a 5-fold cross-validation procedure to minimize over-fitting. In each SVM run, twenty trials of each of four conditions were randomly drawn, without replacement, from all the recorded trials and used to create a population firing rate matrix of $N$ neurons $\times$ 80 trials. Then, in each neuron, the firing rate in each trial was divided by the maximum firing rate of the neuron across the 80 trials. Half of the trials (10 trials from each condition) from each condition were then used to select the parameters with the grid search and subsequently to train the SVM classifier with these parameters. The remaining trials (10 trials from each condition) were then used to test the decoding accuracy after training. The process was repeated 20 times using a different sampling of ten training and ten test trials each time. Based on the classifications, a confusion matrix was created, which indicated the proportion of classifications in which a population firing vector belonging to condition $X$ was classified as condition $Y$.

Our preliminary analysis used firing patterns of 340 neurons recorded during the third post-learning week to test how decoding accuracy changes depending on three parameters: (1) the bin's temporal location relative to CS onset, (2) the size of bin used to construct population firing vectors, and (3) the number of neurons included in a population firing vector. We entered the population firing rate during a series of 200 ms bins over a 1.4 s period around CS onset (50% overlap) to the SVM classifier (*Figure 3—figure supplement 1A*). Decoding accuracy was significantly better than chance even prior to CS presentation, but it further increased at CS onset and remained high until US onset (random permutation test, all data points, $p<0.001$). The high decoding accuracy during CS-US intervals was consistently observed when the input was the population firing patterns with bin sizes of 100 and 50 ms; however, the overall decoding accuracy worsened with a smaller size (*Figure 3—figure supplement 1A*). Next, we used the population firing rate during the first 200 ms after the CS offset to test how decoding accuracy changed depending on the number of neurons included in the analysis (*Figure 3—figure supplement 1B*). Although decoding accuracy improved with a greater number of neurons included in the population firing vector, the classification with vector sizes greater than 150 neurons displayed reliably high decoding accuracy. Therefore, the main analyses were conducted with the population firing vectors during the first 200 ms time window after CS offset of 150 randomly sampled neurons.

To quantify the selectivity for the relational or physical features of the conditions, the same SVM classification procedure was performed after collapsing four conditions into two conditions (for the

**Table 1.** The number of neurons recorded from each rat during each learning stage.

|       | Before | During | Post 1W | Post 2W | Post 3W | Total |
|-------|--------|--------|---------|---------|---------|-------|
| Rat 1 | 42     | 159    | 45      | 56      | 80      | 382   |
| Rat 2 | 64     | 357    | 174     | 162     | 184     | 941   |
| Rat 3 | 78     | 213    | 96      | 77      | 59      | 523   |
| Rat 4 | 42     | 84     | 46      | 46      | 37      | 255   |
| Total | 226    | 813    | 361     | 341     | 360     | 2101  |

relational feature, CS-alone trials and CS-US paired trials; for the physical feature, trials with the ACS and those with the VCS).

Permutation tests were performed for each SVM run using the exact same procedure as above, after assigning, for each population firing vector, a randomized condition label. This procedure, repeated 50 times, each of which generates 40 readouts, yielded the distribution of chance performance of each classifier with 2000 datasets. The raw decoding accuracy was considered as significant when it fell in the 5% upper tail of its corresponding chance performance distribution ($\alpha$ = 0.05). The relative decoding accuracy was defined as the raw decoding accuracy minus the decoding accuracy at the 5% upper tail of its corresponding chance performance distribution. To compare the change in the decoding accuracy across five learning stages, the relative decoding accuracy was calculated in 20 sets of 150 neurons randomly sampled from all recorded neurons in each stage (*Astrand et al., 2014*). The relative decoding accuracy was compared across the stages by one-way ANOVA followed by a posthoc Tukey HSD test.

## Selectivity of single neuron firing

The selectivity of firing responses of single neurons was quantified as the magnitude and consistency of firing differentiation across conditions. The magnitude of firing differentiation was quantified as a differentiation index, which compared mean firing rates during trace intervals between two conditions:

$$\mathrm{Differentiation\ index} = (\mathrm{Fr1} - \mathrm{Fr2})/(\mathrm{Fr1} + \mathrm{Fr2})$$

where Fr1 and Fr2 are averaged firing rates during the CS-US interval across trials in two conditions. For the selectivity for relational features, Fr1 is the mean firing rate during CS-alone trials, and Fr2 was the mean firing rate during CS-US paired trials. For the selectivity for physical features, Fr1 was the mean firing rate during trials with the auditory CS, and Fr2 was the mean firing rate during trials with the visual CS. Raw differentiation indices were converted to absolute values, and these values from all neurons were compared across the five stages of learning with the Kruskal-Wallis test followed by planned pair-wise comparisons with the rank sum test.

The consistency of firing differentiation was quantified as mutual information. It was computed from the joint distributions of firing rates across conditions and takes into account variances across trials within each condition:

$$\sum_{i,j} P(i,j) * log\left(\frac{P(i,j)}{P(i)P(j)}\right)$$

Where *P(i,j)* is the joint probability distribution of condition '*i*' and firing rate '*j*', *P(j)* is the marginal probability distribution of firing rates, averaged across conditions, and *P(i)* is the marginal distribution of firing rate in condition '*i*'. In each neuron, the firing rate was binned into 10 bins to describe the probability distribution. To assess the significance of selectivity, permutation tests were performed for each neuron and for each combination of conditions using the exact same procedure as above, after assigning randomized condition labels to each trial. This procedure, repeated 1000 times, yielded the distribution of the chance level of mutual information values. An observed mutual information value with the correct condition labels was considered as significant when it fell in the 5% upper tail of its corresponding chance distribution. The percentage of neurons with significant mutual information was compared across the stages of learning and over-training by a binomial test. The normalized mutual information was defined as the raw mutual information minus the mean of its corresponding chance distribution, divided by the standard deviation. The values from all neurons were compared across the five stages of learning and over-training with the Kruscal-Wallis test followed by planned pair-wise comparisons with the rank sum test.

## Histology

Upon completion of all recordings, the location of electrodes was marked by electrolytic lesions. Rats were first injected intraperitoneally with an overdose of sodium pentobarbital. For tetrodes, 5 µA was passed through one wire of each tetrode (positive to the electrode, negative to animal ground) for 20 s, for LFP electrodes 20 µA was passed for 45 s. Rats were then perfused intracardially with 0.9% saline followed by 10% buffered formalin. The brain was removed from the skull and

stored in 10% formalin for several days. For cryogenic sectioning, the tissue was infiltrated with 30% sucrose solution, frozen and sectioned in a cryostat (Leica, Wetzlar, Germany) at 50 μm. Sectioned tissue was stained with cresyl violet and imaged under a light microscope to locate electrode locations. Only recordings from tetrodes located in the prelimbic region of mPFC were used for single unit analysis.

## Acknowledgements

The authors thank Drs PW Frankland and M Moscovitch for helpful comments; M Pilkiw and S Sarkar for help with data collection.

## Additional information

### Funding

| Funder | Grant reference number | Author |
| --- | --- | --- |
| Natural Sciences and Engineering Research Council of Canada | Discovery Grant, 2015-05458 | Kaori Takehara-Nishiuchi |
| Canada Foundation for Innovation | Innovation Grant, 25026 | Kaori Takehara-Nishiuchi |
| Natural Sciences and Engineering Research Council of Canada | Canada Graduate Scholarship | Mark D Morrissey |

The funders had no role in study design, data collection and interpretation, or the decision to submit the work for publication.

### Author contributions

MDM, Conceptualization, Data curation, Formal analysis, Funding acquisition, Validation, Investigation, Visualization, Methodology, Writing—original draft, Project administration, Writing—review and editing; NI, Validation, Writing—original draft, Writing—review and editing; KT-N, Conceptualization, Formal analysis, Supervision, Funding acquisition, Writing—original draft, Writing—review and editing, Visualization, Methodology

### Author ORCIDs

Kaori Takehara-Nishiuchi, http://orcid.org/0000-0002-7282-7838

### Ethics

Animal experimentation: All experiments were conducted in accordance with guidelines set forth by the Canadian Council on Animal Care and the Animal Care and Use Committee at the University of Toronto. All protocols were approved by the Animal Care and Use Committee at the University of Toronto (protocol # 20011400).

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
