## [Decision Letter]

Thank you for submitting your article "Prefrontal Coding Underlying Development of Knowledge" for consideration by *eLife*. Your article has been reviewed by two peer reviewers, including John F Disterhoft (Reviewer #2), and the evaluation has been overseen by a Reviewing Editor and Timothy Behrens as the Senior Editor.

The reviewers have discussed the reviews with one another and the Reviewing Editor has drafted this decision to help you prepare a revised submission.

Summary:

In this paper the authors report neural correlates of trace eyeblink conditioning in the mPFC. Neural activity was recorded as rats were trained for several weeks to associated 100ms auditory or visual cues with US. The sessions consisted of blocks of each cue. In each the cue was presented 20x without US and then 80x paired with a US. Anticipatory eyeblink was measured during the 500ms trace interval compared to a pre-CS interval. Elevated eyeblink was considered a CR, and the% of trials with CR's increased during the US blocks but not during the CS-only blocks across sessions. Neural activity was subsequently analyzed via ensemble analysis and was found to distinguish the two visual cues better early in learning than later in learning, whereas the neural activity distinguished the CS-only and CS-US blocks better after learning and during 3 weeks of over-training. This pattern was found to persist if the rats were tested in a novel environment.

Essential revisions:

1) The first and perhaps most significant is that it is not clear that the behavior is conditioned to the cues. It seems like trials on which there was no elevation of responding were discarded. I don't really understand this and what it means for the% CR plotted in the paper. A clearer description of this is necessary. More importantly, it was not stated what the intertrial intervals are. If they are very short, then I think it is possible that the rats are not distinguishing the cue-context associations to guide responding but rather are simply taking their cue from the absence or presence of shock and then responding accordingly. This should be simple to sort out. The authors just need to analyze eyeblinks during a period in the ITI versus their anticipatory period, to each cue, in the two blocks, across sessions, and show that there are the appropriate effects. Specifically, the analysis requires that there be a session X block X pre/post cue interaction and subsequently they should be able to show that this interaction is because with learning, there develops a differential anticipatory response in the US blocks that is not present in the CS-only blocks. This is the same thing that their analysis might be showing, but I am basically not sure. If done, it becomes easier to see why the ensembles don't care about the cues with learning. Indeed, possibly this is the reason. And that may still be fine, but it would be a subtly different point than the authors want to make.

2) Are the authors just showing what we would expect given that the specifics of the cues are all the animals know at the beginning and they are learning the CS-US associations presumably. Wouldn't that lead to exactly the pattern of data that they find? What is different here from what I would expect if I took two cues that are neutral and then paired them with a potent, biologically meaningful US in some contexts but not others? Wouldn't you expect the ensembles to become more highly attuned to the common US predictions that the cues make, and therefore weaker at representing the sensory features of the cues on a relative level? Could the analysis have come out another way? What if you did not train during the subsequent 3 weeks after learning? You mentioned that the transfer to mPFC dependence does not require training right? Isn't it important to show that this "consolidation" does not either? What if it does? Then would it not reflect the process claimed?

3) The learning curves of percent CRs across days shows learning to the paired CS-US trials (Figure 1) in contrast to the low percentage of responses during the block of CS alone trials at the start of both daily epochs. This result actually suggests that the rats learned three relations, the two CS-US contingencies, and that initial trials within each epoch would not be paired with a US. This point could be addressed in the subsection “Rats learned two associative memories sharing a common stimulus relationship”. It might be very interesting to look at the first two trials of each 80 trial block, one might predict that the first trial would be similar to the CS alone trials and that the second trial would exhibit conditioned responding that increases across sessions, i.e. trial 1 of the 80 trial sequence would reinstate previous learning and demonstrate memory. Also in regard to Figure 1, the authors should use a nonparametric analysis given that there are only four rats in the study.

4) Figure 2 is an important summary of data in that it shows the activity for all of the neurons recorded during the learning phase and the Post 3W phase, but several things are not clear and compelling. The data were normalized to the maximum firing rate of the neuron which is unusual for this reader (subsection “Population Firing Rate Matrix”). Most reports indicate either raw firing rates or rates normalized to a time window prior to onset of the CS, i.e. standard scores. One may get more out of the figure if the neurons were ordered by the amount of change relative to baseline, as that would make it easier to understand the responsiveness of the group. It would be good to indicate the number of neurons recorded. The Y axis shows 213 and 80 neurons, but the legend refers to Neuron #340. In the first paragraph of the subsection “Time-dependent changes in the selectivity for relational and physical features were driven by 2 independent changes in single neuron firing properties” it indicates that 2066 neurons were recorded. This probably includes neurons recorded prior to the behavioral plateau – but an explanation should be given somewhere what happened to all of the neurons not illustrated. Finally, the figure lettering in this figure is incorrect – it only includes an A and B section; section C is missing.

5) Figure 5 is shown to indicate changes in the magnitude of response (and consistency of response). A significant difference is indicated for the magnitude (Figure 5), although the change in the index does not appear to increase much relative to the confidence interval. It also seems that the A and D sections of this figure are a bit misleading, as they are schematic diagrams of a "hypothetical change" – and the change illustrated seems to be considerably larger than the actual data in the lower sections of the figure. Perhaps the results would be clearer if plotted as mean ranks rather than the Differentiation Index, especially since the analysis was based on an analysis of rank scores. Similarly, the analysis of mutual information is stated to have been done only on data from the 25th – 75th percentile, the results may change if all values were included.

6) Some discussion of how the authors dealt with the issue of sampling single neurons across the many recording days is in order. Were the tetrodes moved a sufficient amount each day to insure that a separate population of neurons were being recorded from experimental day to experimental day. Are the neurons illustrated in Figure 2 and Figure 5 separable neurons? The general issue of how the individual neurons were defined in Figure 2 and Figure 5 as compared to the total number of neurons reported needs to be discussed.

---

## [Author Response]

*Essential revisions:*

*1) The first and perhaps most significant is that it is not clear that the behavior is conditioned to the cues. It seems like trials on which there was no elevation of responding were discarded. I don't really understand this and what it means for the% CR plotted in the paper. A clearer description of this is necessary.*

During the analysis of eyelid EMG activity, trials were removed when EMG amplitude during the period before CS onset was abnormally large (see new Figure 1—figure supplement 1 for example). The large increase in EMG amplitude was because a rat engaged in grooming, teeth grinding, or climbing. Any sporadic large EMG amplitude before CS onset is problematic for CR detection because CR is defined as an increase in EMG amplitude before US onset relative to the amplitude before CS onset. Further, when this activity is initiated before CS onset, it could carry through to the trace interval and produce false positives. Therefore, these trials were discarded as was done in our previous work (Takehara et al., 2003; Morrissey et al., 2012; Tanninen et al., 2013, 2015; Volle et al., 2016). The proportion of these “hyperactive” trials did not change across sessions in any of four conditions (Figure 1—figure supplement 1). To make this point clear, we added several sentences in the Methods section (subsection “Behavior Analysis”, first paragraph) and two new figures (Figure 1—figure supplement 1).

*More importantly, it was not stated what the intertrial intervals are.*

The intertrial intervals were pseudorandomized across trials and ranged from 20 to 40 seconds. The information has been added to the Results section (subsection “Rats learned two associative memories sharing a common stimulus relationship”).

*If they are very short, then I think it is possible that the rats are not distinguishing the cue-context associations to guide responding but rather are simply taking their cue from the absence or presence of shock and then responding accordingly. This should be simple to sort out. The authors just need to analyze eyeblinks during a period in the ITI versus their anticipatory period, to each cue, in the two blocks, across sessions, and show that there are the appropriate effects. Specifically, the analysis requires that there be a session X block X pre/post cue interaction and subsequently they should be able to show that this interaction is because with learning, there develops a differential anticipatory response in the US blocks that is not present in the CS-only blocks. This is the same thing that their analysis might be showing, but I am basically not sure. If done, it becomes easier to see why the ensembles don't care about the cues with learning. Indeed, possibly this is the reason. And that may still be fine, but it would be a subtly different point than the authors want to make.*

According to the reviewer’s suggestion, we compared the frequency of eyeblink responses during a period before US onset (post-CS phase) against that during a period before CS onset (pre-CS phase). By applying the statistical analysis that the reviewer requested, we confirmed that the frequency of eyeblink responses during the post-CS, but not pre-CS phase became greater across sessions in CS-US paired trials and that this phase-dependent change in eyeblink responses was not observed in CS-alone trials. To make these points clear, we added several sentences in the Results and Methods sections (subsection “Rats learned two associative memories sharing a common stimulus relationship”; subsection “Behavior Analysis”, first paragraph). We also added a new figure that depicts the proportion of trials with eyeblink responses during the pre-CS phase (Figure 1—figure supplement 1).

*2) Are the authors just showing what we would expect given that the specifics of the cues are all the animals know at the beginning and they are learning the CS-US associations presumably. Wouldn't that lead to exactly the pattern of data that they find? What is different here from what I would expect if I took two cues that are neutral and then paired them with a potent, biologically meaningful US in some contexts but not others? Wouldn't you expect the ensembles to become more highly attuned to the common US predictions that the cues make, and therefore weaker at representing the sensory features of the cues on a relative level? Could the analysis have come out another way?*

The key feature of our findings is that the selectivity for physical stimulus features weakened over several weeks *after* the rats had acquired two CS-US associations (Figure 3). If, as the reviewer suggested, the weakened selectivity simply resulted from learning of two CS-US associations, the selectivity for the physical feature should have been weakened as the rats learned the association (i.e. from the “Before” to “Learning” stage in Figure 3). To make this point clear, we have added a sentence to the Discussion (first paragraph).

*What if you did not train during the subsequent 3 weeks after learning? You mentioned that the transfer to mPFC dependence does not require training right? Isn't it important to show that this "consolidation" does not either? What if it does? Then would it not reflect the process claimed?*

This question of learning as a product of “time” versus “experience” underlies the entire field of hippocampal-dependent learning (as compared with non-hippocampal-dependent learning). In the present manuscript, we offer new, qualitative, and quantitative observations about the changes in prefrontal ensemble selectivity taking place over a month of repeated conditionings. Past findings (Takehara-Nishiuchi and McNaughton, 2008; Hattori et al., 2014) demonstrated that some of these changes (CS-US relationship coding) are no less a product of time than experience because they take place with or without repeated conditionings. Although these observations support the hypothesis that changes in prefrontal neuron selectivity are driven by hippocampal-dependent off-line replay, we can not rule-out the possibility that the weakened selectivity for physical features does not involve this process. It is worth noting that the *loss* of selectivity may very well require no active process at all, as it could be accounted for by the lack of reinforcement over time. In that sense, the repeated exposures may be working *against* the effects that we observed. Testing these points directly will require new experiments, ideally experiments which manipulate multiple variables that include the number of different “CS-US” exemplars animals are exposed to, the similarity between exemplars, the temporal proximity between exposures, and elapsed time from the exposures. We have added a section in the Discussion to address these points (fourth paragraph).

*3) The learning curves of percent CRs across days shows learning to the paired CS-US trials (Figure 1) in contrast to the low percentage of responses during the block of CS alone trials at the start of both daily epochs. This result actually suggests that the rats learned three relations, the two CS-US contingencies, and that initial trials within each epoch would not be paired with a US. This point could be addressed in the subsection “Rats learned two associative memories sharing a common stimulus relationship”.*

A sentence on this point has been added to the Results section (end of subsection “Rats learned two associative memories sharing a common stimulus relationship”).

*It might be very interesting to look at the first two trials of each 80 trial block, one might predict that the first trial would be similar to the CS alone trials and that the second trial would exhibit conditioned responding that increases across sessions, i.e. trial 1 of the 80 trial sequence would reinstate previous learning and demonstrate memory.*

According to the reviewer’s suggestion, we examined how ensemble activity in a single trial changed upon the transition from one trial block to the other (new Figure 2 and Figure 2—figure supplement 4). We found that the ensemble activity rapidly evolved into a new, statistically uncorrelated pattern as the block of CS-US paired trials began (Figure 2). During the third week after learning, this transition occurred within the first 10 ACS-US paired trials, which mirrors the time course over which the rats increased the frequency of CR expression at the beginning of CS-US paired trial block (Figure 1). During learning, on the other hand, the transition took ~20 ACS-US paired trials (Figure 2—figure supplement 4). Furthermore, upon the shift from the epoch with the auditory CS to the one with the visual CS, the ensemble similarity abruptly dropped, but it gradually went up when the block of VCS-US paired trials began (Figure 2). These findings provide further support for our view that the mPFC ensemble is selective for the associative stimulus structure and tightly coupled with the behavioral expression of associative memory. To make this point clear, several sentences were added to the Results and Methods sections (subsection “Prefrontal ensembles form codes selective for physical and relational features of memories”; subsection “Population Firing Rate Matrix”, last paragraph) along with two new figures (Figure 2, Figure 2—figure supplement 4).

*Also in regard to Figure 1, the authors should use a nonparametric analysis given that there are only four rats in the study.*

To address the third part of comment 1, we needed to keep ANOVA in the main text; however, we confirmed that the difference in CR% across four conditions was significant with the Friedman test. We included this information in the Results and Methods sections (subsection “Rats learned two associative memories sharing a common stimulus relationship”;).

*4) Figure 2 is an important summary of data in that it shows the activity for all of the neurons recorded during the learning phase and the Post 3W phase, but several things are not clear and compelling. The data were normalized to the maximum firing rate of the neuron which is unusual for this reader (subsection “Population Firing Rate Matrix”). Most reports indicate either raw firing rates or rates normalized to a time window prior to onset of the CS, i.e. standard scores. One may get more out of the figure if the neurons were ordered by the amount of change relative to baseline, as that would make it easier to understand the responsiveness of the group.*

Due to the variation in raw baseline firing rates across neurons, a certain type of normalization was necessary to compare ensemble firing patterns between four conditions. We chose to convert raw firing rates to the ratio to the maximum because it equalizes the dynamic range of firing rate across neurons. This approach also allows for demonstrating the difference in baseline firing rate as well as CS-evoked firing rate across four conditions. The standard score, on the other hand, emphasizes the difference in CS-evoked firings but ignores any across-condition difference in baseline firing rate. The latter is problematic because, as shown in Figure 2, ensemble firing rate before the CS presentation also carried some information about the condition. Therefore, we kept the original figures but added a new section and figure that discusses the ensemble patterns of CS-evoked firings using the standard score (Figure 2—figure supplement 3). As shown, ensemble patterns of CS-evoked firings during the third post-learning week appeared to be more similar for two conditions with the shared relational features than those during learning. This information has been added to the Results and Methods section subsection “Prefrontal ensembles form codes selective for physical and relational features of memories”; subsection 2 Population Firing Rate Matrix”, first paragraph).

*It would be good to indicate the number of neurons recorded. The Y axis shows 213 and 80 neurons, but the legend refers to Neuron #340. In the first paragraph of the subsection “Time-dependent changes in the selectivity for relational and physical features were driven by 2 independent changes in single neuron firing properties” it indicates that 2066 neurons were recorded. This probably includes neurons recorded prior to the behavioral plateau – but an explanation should be given somewhere what happened to all of the neurons not illustrated.*

The manuscript now includes a table (Table 1) that summarizes the number of neurons recorded from each rat during each learning stage. We have corrected all errors on the number of cells in the main text, figure, and figure legends.

*Finally, the figure lettering in this figure is incorrect – it only includes an A and B section; section C is missing.*

We corrected the error.

*5) Figure 5 is shown to indicate changes in the magnitude of response (and consistency of response). A significant difference is indicated for the magnitude (Figure 5), although the change in the index does not appear to increase much relative to the confidence interval. It also seems that the A and D sections of this figure are a bit misleading, as they are schematic diagrams of a "hypothetical change" – and the change illustrated seems to be considerably larger than the actual data in the lower sections of the figure. Perhaps the results would be clearer if plotted as mean ranks rather than the Differentiation Index, especially since the analysis was based on an analysis of rank scores.*

We replaced the original Figure 5 with new figures depicting mean ranks of Differential Index (new Figure 5) or normalized mutual information (new Figure 5). In addition, Figure 5 was replaced with the original Figure 4—figure supplement 1 that showed three actual examples of the distribution of firing rates in two conditions. To reflect these changes, we edited the Results section (subsection “Time-dependent changes in the selectivity for relational and physical features were driven by independent changes in single neuron firing properties”, second and third paragraphs).

*Similarly, the analysis of mutual information is stated to have been done only on data from the 25th – 75th percentile, the results may change if all values were included.*

Although error bars in the original Figure 5 depicted the 25th – 75th percentile, all values were included in the statistical analyses. To make this point clearer, we edited the sentence in the Methods section (Subsection “Selectivity of Single Neuron Firing”).

*6) Some discussion of how the authors dealt with the issue of sampling single neurons across the many recording days is in order. Were the tetrodes moved a sufficient amount each day to insure that a separate population of neurons were being recorded from experimental day to experimental day.*

We did not move tetrodes systematically every day because we needed to record neurons from comparable parts of the prelimbic region across five learning stages. If we had moved tetrodes every day over a month, neurons in the dorsal part of the prelimbic region would have been recorded during early stages of learning, while neurons in a more ventral part of the PrL would have been recorded during later stages. Given the known difference in anatomical characteristics between the dorsal and ventral PrL (e.g., Gabbott et al., 2005), the difference in recording locations across the learning stages becomes an issue: the observed changes in ensemble selectivity may be simply due to the difference in the neuron selectivity between the dorsal and ventral parts of the PrL, rather than changes in the feature selectivity with consolidation.

As the reviewer pointed out, our data appeared to include some neurons recorded across a few days (new Figure 2—figure supplement 1). Having neurons recorded across days was beneficial for the comparison of ensemble selectivity across learning stages because it would reduce variations in sampled neurons across stages. To make this point clear, we added several sentences in the Results and Methods sections (subsection “Prefrontal ensembles form codes selective for physical and relational features of memories”; subsection “Data Preprocessing”) along with a new figure (Figure 2—figure supplement 1).

*Are the neurons illustrated in Figure 2 and Figure 5 separable neurons?*

2A shows the activity of neurons recorded from one of four rats during learning and the third post-learning week. Figure 5 shows the results of analyses that used all neurons recorded from all four rats across all sessions. Therefore, neurons in Figure 2 were a part of the neurons used for the analyses in Figure 5.

If the reviewers were instead referring to the separations between neurons illustrated in Figure 2 and Figure 6 (context selectivity tests), the third post-learning week and context sessions were separated enough in time (seven days on average) that we are confident the populations of neurons recorded in these two sets of sessions are different.

*The general issue of how the individual neurons were defined in Figure 2 and Figure 5 as compared to the total number of neurons reported needs to be discussed.*

An individual neuron was defined as a unit that was well isolated from raw signals recorded on a tetrode (Figure 2—figure supplement 1). Because we minimized the movement of tetrodes across days due to the reason discussed above, some units which appeared to belong to the same neuron were recorded across multiple days (Figure 2—figure supplement 1). We treated each of these units as a separate sample of a neuron. The total number of neurons reported is the summation of the number of isolated units across sessions, tetrodes, and rats. Therefore, our data consist of some neurons repeatedly recorded across days and others recorded once. To make this point clear, we included this information in the Methods section (subsection “Data Preprocessing”).